

# Sulfated polysaccharides from *Phaeodactylum tricornutum*: isolation, structural characteristics, and inhibiting HepG2 growth activity in vitro

Shengfeng Yang, Haitao Wan, Rui Wang and Daijun Hao

Qingdao Tumor Hospital, Qingdao, China

## ABSTRACT

Microalgae, eukaryotic unicellular plants, are increasing in demand due to their use as nutraceutical and food supplements. They consisted different kinds of biologically active components such as polysaccharides. On the other hand, cancer is the leading cause of death globally. At present, there is no efficient method to cure it. Therefore, in this work, we extracted polysaccharides from *Phaeodactylum tricornutum* (PTP), characterized the chemical composition and structure, and investigated its anticancer activity on HepG2 cells. The results showed that PTP was a sulfated polysaccharide with a high Mw of 4,810 kDa, and xylose, fucose, glucose and galactose were the main monosaccharides. PTP has significant anticancer activity in a dose-dependent manner (up to 60.37% at 250 ug/mL) according to MTT assays. Furthermore, cycle analysis was carried out to explain its anticancer activity. The results showed that it exhibited anticancer effect mainly through the induction of apoptosis without affecting the cycle and mitosis of HepG2 cells. This might make it a potential drug for anticancer treatment in the future.

## INTRODUCTION

Microalgae are promising source of biomass due to their advantageous features such as their phototropic nature, high growth rate, lack of competition with food crops for arable land, and abundant nutritious components, such as protein, pigments, and trace elements (*Hamilton et al., 2014*; *Wang et al., 2015*). Therefore, it has been used as feedstock, such as in food, feed, functional foods, biofuels, or chemicals integrated in novel biorefinery concepts (*Zhu, 2015*; *Vandamme et al., 2018*). Unlike terrestrial plants, the biologically active compounds extracted from microalgae have shown unique properties, such as antibacterial, antiviral, antifungal, antioxidative, anti-inflammatory, and anti-tumor properties (*De Jesus Raposo, de Morais & de Morais, 2013*; *Guzmán et al., 2003*; *Hayashi et al., 1996*; *Kaji et al., 2004*; *Challouf et al., 2011*; *De Jesus Raposo, de Morais & de Morais, 2015*). From economical point of view, polysaccharides from algae are promising products due to their abundance in algae (*Kraan, 2012*). Polysaccharides can be

Corresponding authors
Shengfeng Yang,
xiaolinchen_79@163.com
Daijun Hao, haodaijunzhl@163.com

extracted from algae by several "green" extraction techniques, such as microwave-assisted extraction (*Rodriguez-Jasso et al., 2011*) and enzyme-assisted extraction methods (*Ko et al., 2012*). The characteristics of different polysaccharides from microalgae, including their composition and structure, were discussed (*De Jesus Raposo, de Morais & de Morais, 2013*). It was reported that *G. impudicum* and *C. vulgaris* contained homo galactose (*Yim et al., 2007*) and glucose (*Nomoto et al., 1983*), respectively. However, the other polysaccharides from microalgae are heteropolymers of galactose, xylose, glucose, rhamnose, fucose, and fructose (*Matsui et al., 2003*; *Talyshinsky, Souprun & Huleihel, 2002*; *Raposo, Morais & Morais, 2014*). *Ford & Percival (1965)* found that the structure of the polysaccharides from *Phaeodactylum tricornutum* was a ramified sulfated flucoronomannan, with a backbone composed of β-(1,3)-linked mannose. Many studies have shown that the polysaccharides from microalgae are characterized by antibacterial, antitumor, and antiviral properties (*Michalak & Chojnacka, 2015*).

As a kind of diatom, PTP has been found in great abundance in coastal and oceanic waters (*Bautista-Chamizo et al., 2018*). It contains approximately 36.4% crude protein, 26.1% carbohydrate, 18.0% lipid, 15.9% ash, and 0.25% neutral detergent fiber on a dry weight (dw) basis (*Rebolloso-Fuentes et al., 2001*). In addition, it can accumulate valuable pigments such as fucoxanthin, triacylglycerols, and omega-3 long-chain polyunsaturated fatty acids, such as eicosapentaenoic acid (EPA; C20:5) (*Kim et al., 2012*; *Ryckebosch et al., 2012*; *Yu et al., 2009*; *McClure et al., 2018*). Currently, it is commercialized for its lipids, especially EPA, and several studies have sought to increase the production yield of EPA and biomass (*Grima et al., 1994*; *Alías et al., 2004*; *McClure et al., 2018*). In recent years, due to its many therapeutic activities, fucoxanthin has been commercialized from PTP. However, there is little research about the polysaccharides extracted from PTP. Therefore, to make full use of the alga, in this paper, we extracted polysaccharides from PTP, characterized its chemical structure, and studied the anticancer activity of the polysaccharides.

## MATERIALS AND METHODS

### *Phaeodactylum tricornutum* samples and reagents

Dried PTP powder was supplied by the Institute of Oceanology, Chinese Academy of Sciences. All the reagents used were of analytical grade and commercially available unless otherwise stated.

### Extraction of polysaccharides from *Phaeodactylum tricornutum* (PTP)

The extraction diagram was as Fig. 1.

The dried PTP powder was extracted by the Soxhlet method with ethanol to remove pigments and lipids. The residue was then dried in an oven at 50 °C, and polysaccharides were extracted by hot distilled water with the assistance of ultrasonic methods. The optimal temperature, times of ultrasonic treatment, and extraction time were determined (shown in Supplemental Information). According to the optimal conditions, the residue algal powder was treated by the ultrasonic method for 20 times, 10 s working,

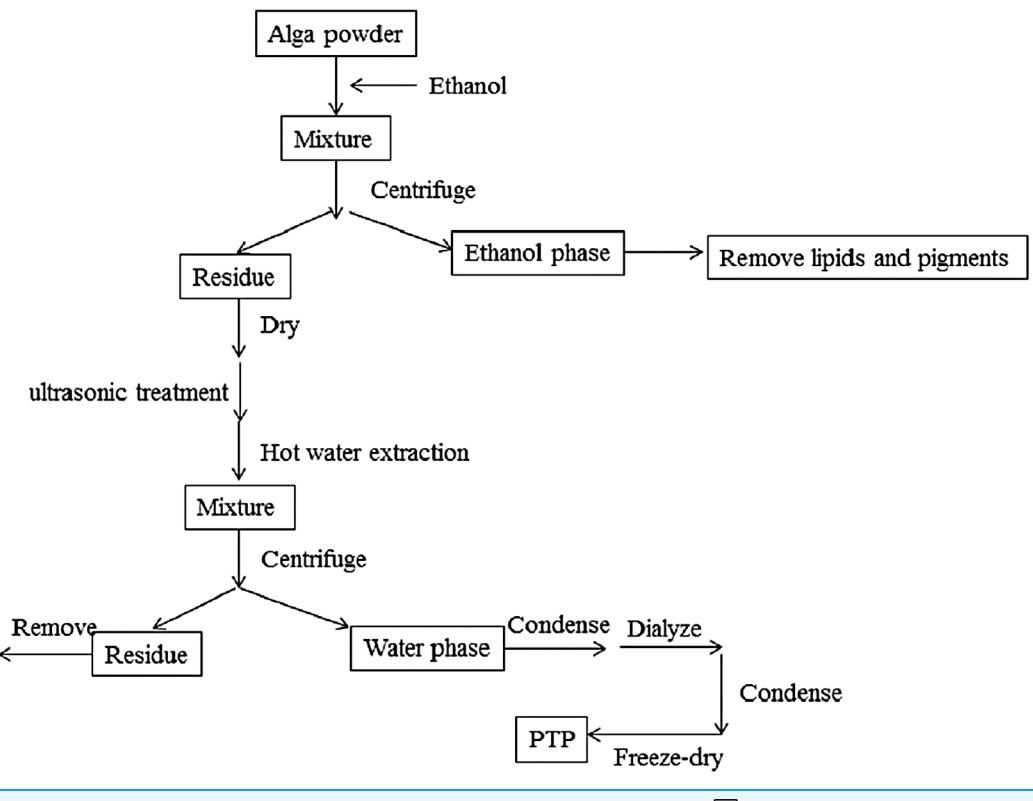

**Figure 1 Extraction process.** The extraction diagram of PTP.

10 s rest, and 380 W power. Then, it was extracted at 80 °C for 2 h with stirring. The solution produced by filtration was condensed by rotary evaporator and dialyzed for salt removal. The obtained solution was condensed again, and the final solution was freeze-dried to get the purified sulfated polysaccharides, called PTP.

## Chemical characterization

The Mw of PTP was measured by HPLC with a TSK gel G4000PWxl column using 0.05 mol/L $Na_2SO_4$ as the mobile phase on an Agilent 1260 HPLC system equipped with a refractive index detector. The column temperature was 35 °C, and the flow rate of the mobile phase was 0.5 mL/min. Dextran standards with a Mw of 1, 5, 12, 50, 80, 270, and 670 KDa (Sigma, Mendota Heights, MN, USA) were used to calibrate the column.

Total sugars were analyzed by the phenol-sulfuric acid method (*Dubois et al., 1956*) using galactose as the standard. sulfated content was determined by the barium chloride gelatin method (*Kawai, Seno & Anno, 1969*). The molar ratios of the monosaccharide composition were determined according to *Sun et al. (2017)*. 1-phenyl-3-methyl-5-pyrazolone pre-column derivation HPLC was used to determine the molar ratio of the monosaccharide composition. Briefly, 10 mg polysaccharide sample was dissolved into one mL distilled water. The mixture was hydrolyzed in 4 mol/L trifluoroacetic acid, followed by neutralization with sodium hydroxide. Then, HPLC was used to determine every monosaccharide composition on a YMC Pack ODS AQ column (4.6 mm × 250 mm). Mannose, rhamnose, fucose, galactose, xylose, glucose, and

glucuronic acid from Sigma-Aldrich were used as standards. FT-IR spectra of PTP were determined on a Nicolet-360 FT-IR spectrometer between 400 and 4,000 cm$^{-1}$.

## Evaluation of inhibiting HepG2 growth activity in vitro

### Cell culture

HepG2 cells purchased from Kunming Cell Bank, Chinese Academy of Sciences, were cultured in DMEM supplemented with 10% fetal bovine serum solution, 100 U/mL penicillin and 100 mg/mL streptomycin at 37 °C in a humidified atmosphere containing 5% $CO_2$.

### Evaluation of inhibiting HepG2 growth activity in vitro

The cell growth inhibitory activity of PTP with different concentrations (50, 100, 150, 200, and 250 ug/mL) was assessed by MTT assay. The cells were seeded in a 96-well plate at a concentration of $1 \times 10^4$ cells/mL and incubated with various concentrations PTP for 48 h. Then, 200 ul 0.5 mg/mL MTT solution was added to each well. After 4 h incubation, the plates were centrifuged for 10 min at 8,000 rpm. MTT solution was removed. And 200 uL DMSO was added into each well. The absorbance at 570 nm was determined.

### Apoptosis assessment

The apoptosis states of HepG2 cells were determined by an Annexin V-FITC/PI apoptosis kit. Cells were collected and washed with ice-cold PBS twice. Then, the cells were resuspended and diluted to $1 \times 10^6$ cell/mL with binding buffer. The suspended cells were dyed by 10 μL of Annexin V-FITC for 30 min at room temperature and then stained with five μL of propidium iodide (PI) for 5 min. After incubation, the apoptosis of cells was determined by flow cytometry with Guava® easyCyte 6-2L (Millipore, Billerica, MA, USA).

### Analysis of the cell cycle

A cell cycle analysis kit (Beyotime, Haimen, Jiangsu, China) was used to analyze the cell cycle according to the manufacturer's instructions. Briefly, cells were plated in DMEM with different concentrations of sample for 24 h. Then, both the suspension and the adherent cells were collected and placed into the flow cytometry tube and centrifuged at 1,500 rpm for 5 min to obtain cell pellets. After that, the cell pellets were washed with precooling PBS and fixed in ice-cold 70% ethanol overnight at 4 °C. Fixed cells were rewashed with PBS and incubated with PI staining solution (0.5 mL of staining buffer, 25 μL of PI staining solution, and 10 μL of RNAase A) for 30 min at 37 °C in the dark. Cell cycle analysis was carried out with Guava® easyCyte 6-2L (Millipore, Billerica, MA, USA) using 10,000 counts per sample. The percentage of cells distributed in the different phases of G0/G1, S, and G2/M were recorded and analyzed.

## Statistical analysis

All data are shown as means ± SD (standard deviation) of three independent experiments to ensure the reproducibility of the results. Statistical analysis was performed using SPSS. The difference among groups was analyzed by one-way ANOVA.

**Table 1 Chemical composition.**

| Sample | Total sugar/% | Sulfate/% | Mw/kDa | Monosaccharides composition (Molar ratio) | | | | | | |
|---|---|---|---|---|---|---|---|---|---|---|
| | | | | Man | Rha | Glc A | Glc | Gal | Xyl | Fuc |
| PTP | 29.94 | 20.36 | 4810 | 0.00 | 0.25 | 0.68 | 0.53 | 0.56 | 1.00 | 0.75 |

Notes:
Chemical composition of PTP (%w/w dry weight).
Man, mannose; Rha, rhamnose; Glc A, glucuronic acid; Gal, galactose; Glc, glucose; Xyl, xylose; Fuc, fucose.

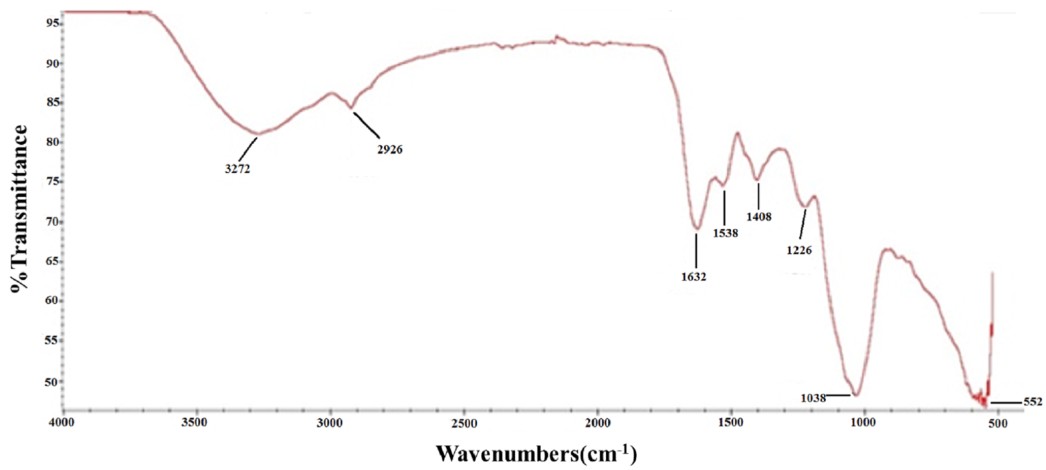

**Figure 2 FTIR.** FT-IR spectra of PTP.

# RESULTS

## Chemical characterization

*Phaeodactylum tricornutum* was extracted and purified from , with a yield of 1.5%(% dw). It was further characterized regarding Mw, total sugars, sulfate content, and monosaccharide composition (Table 1).

According to Table 1, the total sugar and sulfate contents were 29.94% and 20.36%, respectively, which indicated that PTP was a type of sulfated polysaccharide. The Mw of PTP was higher (4,810 kDa). The results of the monosaccharide composition showed that the most common monosaccharide of PTP was xylose, followed by fucose, glucose, and galactose, with a small amount of rhamnose. The glucuronic acid content of PTP (0.68) was higher. These results indicated that PTP was a hybrid and acidic polysaccharide.

To further characterize the chemical structure of PTP, the corresponding FT-IR spectrum was examined (Fig. 2). The O–H stretching vibration appeared at 3,272 cm$^{-1}$, and the C–H stretching vibration appeared at 2,926 cm$^{-1}$. The adsorption at 1,632 and 1,408 cm$^{-1}$ represented the asymmetric and symmetric stretching vibration of C=O, respectively. The adsorption at 1,226 and 1,038 cm$^{-1}$ corresponded to the S=O stretching vibration and C–O–H deformation vibration, respectively. These results further indicated that PTP was an acidic and sulfated polysaccharide, which chelated with other positive ions.

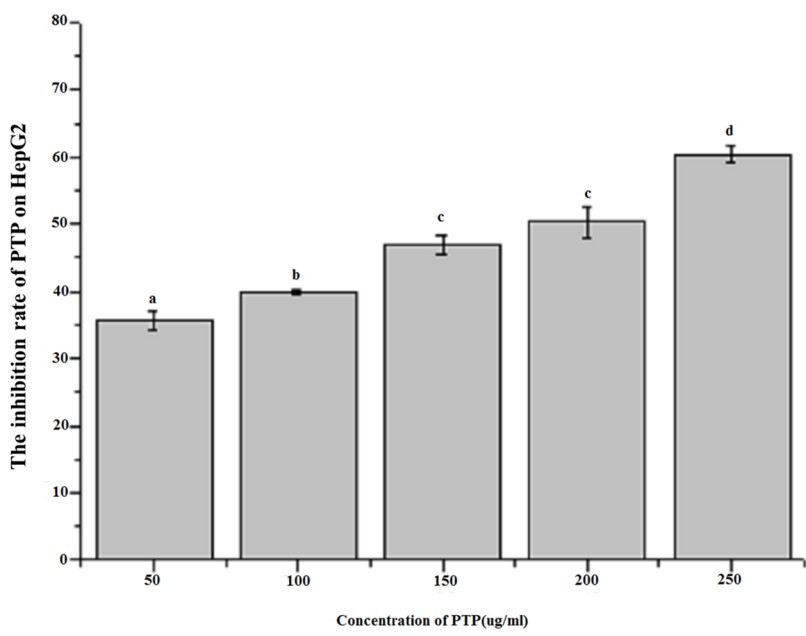

**Figure 3 Inhibition rate of HepG2 by MTT assay.** The effect of different concentration PTP on the inhibition rate of HepG2 by MTT assay for 48h.               

## Evaluation of inhibiting HepG2 growth activity in vitro

Figure 3 shows the inhibitory effect of different concentrations of PTP on HepG2 tumor cells. The results indicated that PTP had an antiproliferative effect on HepG2 cells in a dose-dependent manner. With concentration increasing, PTP had higher inhibitory activity, and the inhibition rate was up to 60.37% when the concentration was 250 ug/mL. However, the manner of PTP inhibiting HepG2 growth was not clear. To analyze the main cause, we determined the cell apoptosis and cell cycle by flow cytometry.

## Induction of apoptosis according to cell cycle analysis

Figure 4 shows the results of flow cytometry, when the HepG2 cells were treated with different concentrations of PTP. From the results, we deduced the apoptosis rate under different concentrations of PTP (shown in Fig. 4). From Fig. 4, when HepG2 cells were treated with PTP, the apoptosis rate increased in a dose-dependent manner, although it decreased slightly under 200 ug/mL PTP. When the concentration of PTP was 250 ug/mL, 30% apoptosis of cells was induced. Double negative PI-Annexin V cells accounted for about 63%. The above results were consistent with those of the MTT assay. They indicated that PTP could significantly induce cell apoptosis. Then, we determined the HepG2 cell cycle rate under three different concentrations (50, 150, and 250 ug/mL) of PTP, as shown in Fig. 5. From Fig. 5, the treatment of different concentrations of PTP did not influence the HepG2 cell cycle rate, which might indicate that PTP's anticancer effect occurred mainly through induction of apoptosis without affecting the mitosis of HepG2 cells.

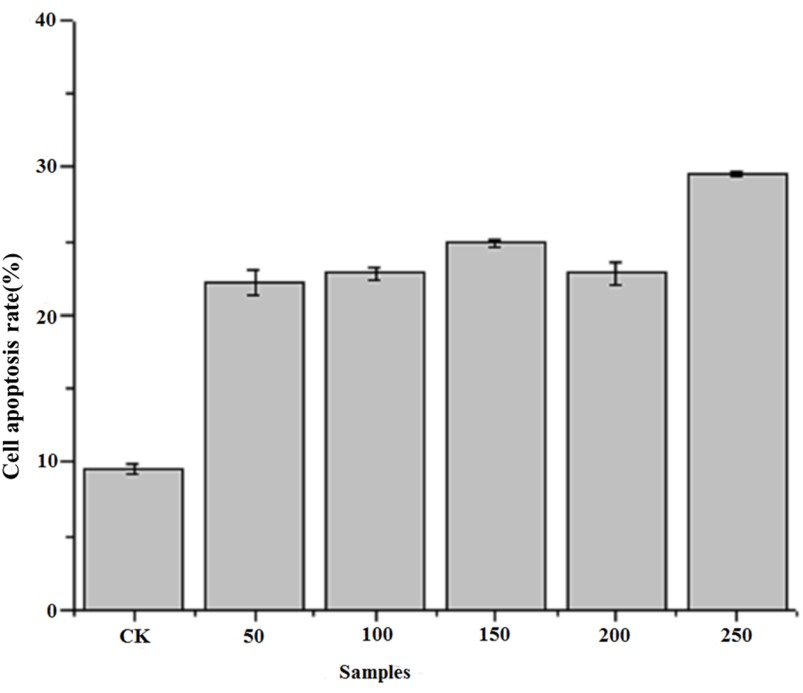

**Figure 4 Cell apoptosis rate.** The cell apoptosis rate under different concentration of PTP.

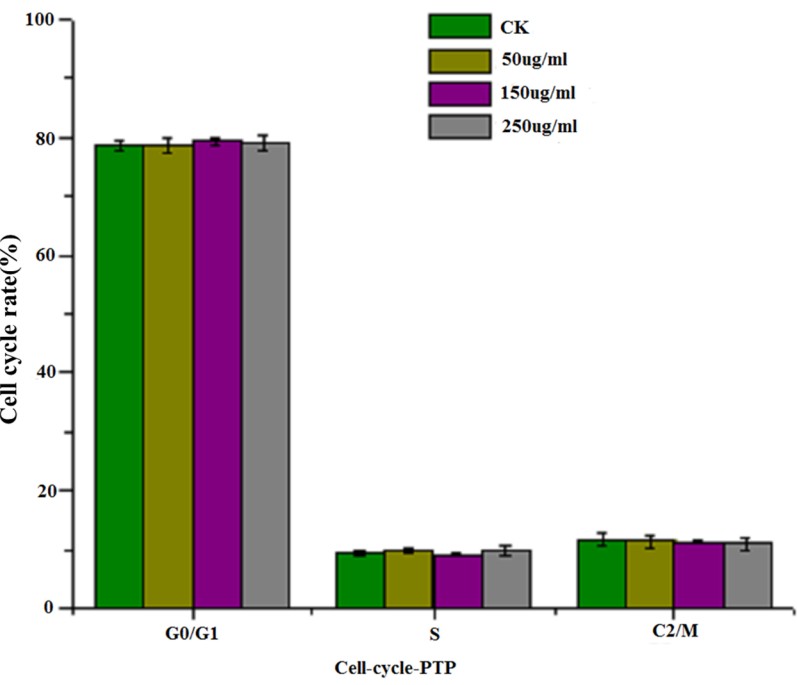

**Figure 5 Cell cycle.** The cell cycle rate under different concentration of PTP.

## DISCUSSION

Cancer is the leading threat to the world population, and it is the first-leading cause of death worldwide. The current cancer treatments often cause side effects (*Samarakoon et al., 2014*; *Boopathy & Kathiresan, 2010*). Recently, due to their favorable properties, polysaccharides from microalgae have been given increased attention. Polysaccharides from *Spirulina platensis* have been shown to have antitumor functions on human HT-29 cells (*Xu et al., 2012a*), MB-231 cells (*Xu et al., 2012b*), HeLa cells (*Yu & Zhang, 2003*) and HepG2 cells (*Di & Wang, 2013*). Polysaccharides from *Platymonas subcondigoramis* inhibited melanoma (*Liu et al., 2007*). *Umemura et al. (2003)* showed that polysaccharides from the dinoflagellate *Gymnodinium* sp. exhibited significant cytotoxicity against a variety of cancer cells, which meant that the polysaccharides might be a potential anticancer chemotherapeutic agent. For PTP, some references reported antioxidant (*Agustini & Kusmiati, 2017*), anti-obesity (*Kim et al., 2016*), anti-inflammatory, and immunomodulatory activities (*Kim et al., 2013*; *Guzmán et al., 2003*). A novel fatty alcohol ester isolated from PTP showed apoptotic anticancer activity (*Samarakoon et al., 2014*). Few studies have addressed polysaccharides isolated from PTP. *Agustini & Kusmiati (2017)* extracted endo-exopolysaccharide and determined its antioxidant activity on DPPH. The composition of the polysaccharides included xylose, glucose, and galactose. Similarly, the monosaccharide composition of PTP mainly included xylose, fucose, glucose and galactose. Fucose was not reported by *Agustini & Kusmiati (2017)*, which may be due to the different origins of the algae. In this paper, we determined not only the monosaccharide composition but also the total sugar, sulfate contents and Mw (29.94%, 20.36%, and 4,810 kDa, respectively) and found that PTP is a complicated sulfated polysaccharide. A type of lipopolysaccharide extracted from Phaeodactylum tricornutum exhibited anti-inflammatory activity by blocking the activation of nuclear factor-κB and phosphorylation of p38 mitogen-activated protein kinases, extracellular signal-regulated kinases 1 and 2 and c-Jun N-terminal kinase (*Kim et al., 2013*). However, there was no related information about the lipopolysaccharide. To our knowledge, no anticancer activity has been reported for PTP. In this paper, we determined the anticancer activity of PTP on HepG2 cells. Significant anticancer activity (up to 60.37% under 250 ug/mL) by MTT assays, which was much better than for polysaccharides isolated from *Spirulina platensis* (*Xu et al., 2012a*, *2012b*).

In addition, several studies reported that polysaccharides isolated from *Spirulina platensis* exhibited anticancer activity by blocking G0/G1 phase of cancer cells, which induced the mitosis of cancer cells and led to apoptosis of the cells (*Xu et al., 2012a*, *2012b*; *Yu & Zhang, 2003*; *Di & Wang, 2013*). However, in this paper, although the apoptosis rate of HepG2 cells increased, cell cycle analysis indicated that PTP's anticancer effect occurred mainly through induction of apoptosis without affecting the cell cycle and mitosis of HepG2 cells. This result might differ according to the chemical components and structure. It needs further investigation. In addition, some references reported that microalgae polysaccharides have the capacity to modulate the immune system so that they

display anticancer activity in vivo (*Andrade et al., 2018*). For this study, only in vitro cell experiments were carried out, and it is necessary to explore the anticancer activity in vivo. Further research will address this issue.

## CONCLUSION

In this paper, a sulfated polysaccharide (PTP) was extracted from PTP with a high Mw (4,810 kDa). The monosaccharide composition of PTP was mainly xylose, fucose, glucose, and galactose. MTT assays showed that PTP has significant anticancer activity (up to 60.37% under 250 ug/mL). Furthermore, the anticancer effect occurred mainly through induction of apoptosis without affecting the cell cycle and mitosis of HepG2 cells. Thus, PTP may be a potential drug for anticancer treatment.

### Funding

This study was supported by a research grant from the Key Laboratory of Natural Product Chemical Biology, Ministry of Education, Shandong University (CB-201708). The funders had no role in study design, data collection and analysis, decision to publish, or preparation of the manuscript.

### Grant Disclosures

The following grant information was disclosed by the authors:
Key Laboratory of Natural Product Chemical Biology, Ministry of Education, Shandong University: CB-201708.

### Competing Interests

The authors declare that they have no competing interests.

### Author Contributions

- Shengfeng Yang conceived and designed the experiments, performed the experiments, prepared figures and/or tables, authored or reviewed drafts of the paper, approved the final draft.
- Haitao Wan performed the experiments.
- Rui Wang analyzed the data.
- Daijun Hao analyzed the data, contributed reagents/materials/analysis tools.

### Data Availability

The raw measurements are available in the Supplementary Files.

### Supplemental Information

Supplemental information for this article can be found online at http://dx.doi.org/10.7717/peerj.6409#supplemental-information.

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
