# Peer review of "Sulfated polysaccharides from Phaeodactylum tricornutum: isolation, structural characteristics, and inhibiting HepG2 growth activity in vitro"

_PeerJ, doi:10.7717/peerj.6409_

## Round 0.1 · original submission · Major Revisions

It is my opinion as the Academic Editor for your article - Sulfated polysaccharides from Phaeodactylum tricornutum: isolation, structural characteristics, and anticancer activity - that it requires a number of Major Revisions.

Reviewer 1 ·

Basic reporting

This article does a wonderful job of isolating and testing the efficacy of sulfated polysaccharides from PTP. The manuscript is well written albeit a minor fixing and would be easily assessable to the international audience. The authors did a commendable job in introducing the wide spread usage of Microalgae citing literature and the different uses of PTP. Figures are relevant but lack labeling, legend and description. (See my comments for the author)

Experimental design

The research is original and clarifies the aims and studies the role of Polysacs from PTP. The methods section seems to be incomplete and lacks a lot of material for instance MTT assay, cell numbers used for the experiment, time points assessed which would make replication impossible. (See my comments for the author). The most important issue I have here is the missing time-course of the effect of the drug. (See my comments for the author). I also have the question as to why only 1 cell line was tested for drug efficacy (HCC). It would be nice if the authors can replicate the same results in another cancer type.

Validity of the findings

The data is robust and proper statistical methods were used. The conclusions for the most part are well stated.

Additional comments

The manuscript is well written and is easy to follow. It has a concise methodology and results are well presented and adhere to the standards of Peer J. However, there are some issues that need to be clarified, a few experiments performed before acceptance. Please see my comments below:-
Methods and Experimental Design:-

1] I would suggest the authors perform a time course experiment to determine for the efficacy of PTP. (Day 0, Day 1,3 and 5) along with the PTP concentrations mentioned here. Also, it would be good if you show how PTP compares to No treatment overtime and see the uninhibited growth rate of HepG2 without any perturbation. Methods section does not even mention about the MTT assay at all, I strongly suggest the authors add a description about the MTT assay. I would also suggest the authors describe all the information like number of cells used, time points assessed and analyzed.

2] Why was HepG2 chosen as the model cell line for the manuscript? I would strongly suggest that the authors repeat these experiments with different cell lines of different cancer types and show that PTP indeed has anticancer activity and is not limited to HepG2. Again a time course study would be welcomed here. I would also suggest toning down the claim of anticancer activity based on the current findings from 1 cell line or be specific about HepG2 or Hepatocellular carcinoma if we can reproduce this effect in other HCC cell lines.

Results:-
1] Titles for results can be made more relevant so as to give the gist of the result which can be expanded upon in the description. For instance, “PTP inhibits HepG2 growth invivo”.

2] Line 147 to 150 is confusing and contradicting. The current phrasing makes it difficult to comprehend and is convulative. I suggest stating your findings in a more succinct manner.

3] I would suggest the authors expand upon the findings in the Section 3.3 “Cell Cycle Analysis” instead of statements like from the result we deduced…. (line 162 -172). In the current format, the authors leave the interpretation to the readers.

Figures:-
1] All the figures lack a description. Figure 1 lacks Y-axis information and description.

2] Figure 2 lacks description

3] Figure 3 lacks description, axis information ( Please indicate what does red and green colors represent)

Reviewer 2 ·

Basic reporting

The manuscript peerj-31007 reported isolation, structural characteristics, and
anticancer activity of sulfated polysaccharides from Phaeodactylum tricornutum.
The subject of this study meets the scope of PeerJ . The article has been carefully reviewed and I come up to the conclusion that the acceptance of this contribution may be considered after major revision.

Experimental design

Only specific comments about characterization (Mw).

Validity of the findings

Conclusions should be more straightforward. Therefore, please re-write this section to be more effective.

Additional comments

The manuscript peerj-31007 reported isolation, structural characteristics, and
anticancer activity of sulfated polysaccharides from Phaeodactylum tricornutum.
The subject of this study meets the scope of PeerJ . The article has been carefully reviewed and I come up to the conclusion that the acceptance of this contribution may be considered after major revision.
Specific Comments:
1. Molecular weight (Mw) determination: The experiment described corresponds to a molecular weight distribution determination and the standards used were in the range 1-670 kDa while the estimated Mw of the sample was 4810 kDa. It is necessary to use standards presenting Mw in the range of the sample analyzed.
2. In Table 2, chemical composition (sugars and sulfate) represent around 50% of the sample. What is the other 50%? Was protein content determined? Minerals?
3. Neutral sugars quantification is reported ‘according to Sun YH, Chen XL, Cheng ZQ, Liu S, Yu HH, Wang XQ and Li PC. 2017. Degradation of
Polysaccharides from Grateloupia filicina and their antiviral activity to avian Leucosis virus subgroup J. Marine Drugs 15, 345’. However, it is necessary a more detailed description of the method used.
4. In general, Figures resolution need to be improved.
5. In Figure 1, FT-IR spectrum y-axis corresponds to % Transmittance?
6. In Figures 3, 4 and 5 what does it means CK?
7. In general, Figure captions need to be improved.

Reviewer 3 ·

Basic reporting

Background and Literature is nicely written.

Experimental design

Experimental design is good.

Need more information-as mentioned in comments below.

Validity of the findings

Conclusions need further data to support.

Additional comments

Comments on manuscript # 31007 "Sulfated polysaccharides from Phaeodactylum tricornutum: isolation, structural characteristics, and anticancer activity" by Yang et. al.
The author studied the sulfated polysaccharide from algae-

Following things need to be explained in the MS.
1. Result 3.1 section, author mentioned characterization in table 1: Author didn’t show purification product after HPLC, which is important to make clear the starting material is perfect. Should be included as a Fig. before characterization?

2. M&M heading, Extraction section: Author used ultrasonication to extract the polysaccharide and didn’t showed optimization parameters, saying length limit of paper (line 71-72)-I wonder why author thought that is not important? Quality control parameters are important always to reproduce the technique-should be included as supplementary data.

3. Author characterized anti-cancer property against HEPG2 cell line and showed 60% at 250 ug/mL. In FT-IR results author mentioned that this polysaccharide is acidic in nature (line 137)-Did the author checked the pH of the culture medium after addition of the polysaccharide? Low pH will also kill the cells.

4. Or is it cell lysis? Cell imaging will also show does it lyse the cells (like HEPG2).

5. FTIR results, polysaccharide is sulfated in nature (same line 137)-Did the author checked any anti-oxidant property? What happens to ROS after addition to HEPG2 cell line, will give the clue about the mechanism of action?

6. Author didn’t check the stability of the polysaccharide in the culture media. Stability Should be checked? For how long the HEPG2 cells were treated? Should also be included in Fig 2.

7. Line 166-normal cells accounted for about 63%-It is a wrong sentence-Should be corrected by saying double negative PI-Annexin V cells.

8. Figure 3 and 4 are same, one is sufficient.

9. Purification flow diagram can also be good to represent the technique of extraction.

The paper is technically sound and well written in nice English. Comments above- will make the MS better.

---

## Round 0.2 · Major Revisions

Thank you for your submission to PeerJ! Please address the remaining comments from Reviewer 3.

Reviewer 1 ·

Basic reporting

The authors made the necessary changes and the manuscript is ready to be published.

Experimental design

No comment

Validity of the findings

No comment

Additional comments

No comment

Reviewer 2 ·

Basic reporting

In general the manuscript has been improved.

Experimental design

The methods are well described in this version.

Validity of the findings

The discussion and conclusion sections have been improved.

Additional comments

All the comments made by the reviewer are well addressed by the authors. So I would suggest to consider for its publication.

Reviewer 3 ·

Basic reporting

Poor English-Should be proof read by all authors-see comments.

Experimental design

Experimental design can be improved-see comments

Validity of the findings

Conclusions need further experimental validation-see comments.

Additional comments

Comments on manuscript # 31007v1 "Sulfated polysaccharides from Phaeodactylum tricornutum: isolation, structural characteristics, and inhibiting HepG2 growth activity in vitro" by Yang et. al.

My following comments last time-they are still unanswered

I believe following things need to be explained in the MS for the betterment of science.
1. Result 3.1 section, author mentioned characterization in table 1: Author didn’t show purification product after HPLC, which is important to make clear the starting material is perfect. Should be included as a Fig. before characterization?

Author response: In this manuscript, we did not use HPLC to purify the polysaccharide. We used HPLC to determine monosaccharide composition.
In revised MS, Line 87: Author wrote Mw was determined by HPLC- the chromatogram should be included in MS in order to show how much purity is in the starting material?

2. M&M heading, Extraction section: Author used ultrasonication to extract the polysaccharide and didn’t showed optimization parameters, saying length limit of paper (line 71-72)-I wonder why author thought that is not important? Quality control parameters are important always to reproduce the technique-should be included as supplementary data.

Author response: We would supply these parameters as supplementary data.
Author provided these parameters that is nice.

3. Author characterized anti-cancer property against HEPG2 cell line and showed 60% at 250 ug/mL. In FT-IR results author mentioned that this polysaccharide is acidic in nature (line 137)-Did the author checked the pH of the culture medium after addition of the polysaccharide? Low pH will also kill the cells.

Author response: Although PTP is acidic, it is chelated with other positive cations, so the pH is about 8.0. And the condition cannot kill the cells.
Author concluded by chemical characterization (line 159 in revised MS) and FTIR (line 166 in revised MS), that it is acidic in nature-By science that means overall charge is acidic? Otherwise author should mention in MS that acidic charge is neutralized by others and overall charge is?????

4. Or is it cell lysis? Cell imaging will also show does it lyse the cells (like HEPG2)

Author response: In our opinion, the conditions and the characteristics of PTP did not make the cell lyse, although cell imaging was not done.
Cell imaging is easy to highlight what this PTP is doing to whole cells?

5. FTIR results, polysaccharide is sulfated in nature (same line 137)-Did the author checked any anti-oxidant property? What happens to ROS after addition to HEPG2 cell line, will give the clue about the mechanism of action?

Author response: We did not check the antioxidant property of the polysaccharide. Maybe, just like other algae polysaccharides, PTP might have antioxidant property and we can do separate experiments to certify it. However, in culture media, it is difficult to determine the antioxidant property and it is not our emphasis.

For authors Clarity-Check ROS after PTP treatment vs no treatment? Include in MS.

6. Author didn’t check the stability of the polysaccharide in the culture media. Stability Should be checked? For how long the HEPG2 cells were treated? Should also be included in Fig 2.

Author response: The treatment time has been added in Fig.2. Although we did not check the stability of the polysaccharides in the culture media, according to our experience on polysaccharides study, the polysaccharide was stable in the culture media unlike protein. And there is other paper about the polysaccharide anticancer activity( Samarakoon et al., 2014) which did not show the stability of the polysaccharide.

Whether other studies included stability or not is not the proper answer. Pre-incubation in complete culture media (for treatment time period) before adding onto the cell culture vs freshly added one will tell the stability-simple.


New comments:
7. Line 151: Author mentioned Yield is 1.5%- First author should mention that yield is written w.r.t. what, initial dry weight or something else? Second that represents a poor experimental procedure for Extraction (nobody is going to use that)-Did author optimized the extraction method, is the best one used (or given in revised MS) finally? These should be clearly written in revised MS, that is why in comment 2, technique parameters are important to highlight your experimental procedure in order to reproduce those results.

8. Line 174: While, what’s the manner of PTP inhibiting HepG2 growth? This was not clear-should be rephrased in a good English.

9. Line 216: There was no other information about polysaccharide in this reference-Should be connected to the previous line sentence of Ref Agustini, otherwise no meaning. Connection between sentences is missing.

Poorly written English. MS should be proof-read by all authors which is lacking in revised MS.
Comments above- will make the MS better.

I recommend the MS “acceptance for publication with major revision”.

---

## Round 0.3 · accepted · Accept

I am pleased to inform you that your article has been accepted for publication, subject to any revisions that might be required by the production group.

# Reviewer 3 ·

Basic reporting

Written in standard English.

Experimental design

Experimental methodology well written.

Validity of the findings

Conclusions well stated.

Additional comments

Chromatogram should be well labeled (what each peak shows???) and included in MS.
Flow cytometry results for cell lysis should also be included in MS.